# Regional Differentiation and Influencing Factor Analysis of Residents’ Psychological Status during the Early Stage of the COVID-19 Pandemic in South China

**DOI:** 10.3390/ijerph182211995

**Published:** 2021-11-15

**Authors:** Yeqing Cheng, Yan Chen, Bing Xue, Jinping Zhang

**Affiliations:** 1College of Geography and Environmental Sciences, Hainan Normal University, Haikou 571158, China; HSchengyq@hainnu.edu.cn (Y.C.); maryzhjp@126.com (J.Z.); 2Institute of Applied Ecology, Chinese Academy of Sciences, Shenyang 110016, China; xuebing@iae.ac.cn

**Keywords:** psychological status, regional differentiation, COVID-19 pandemic, multiple logistic regression methods, South China

## Abstract

A scientific understanding of the impact of COVID-19 on the psychological status of residents is important for improving medical services and responding to public health emergencies. With the help of some of the most popular network communication tools (including Wechat and Weiboand QQ), online questionnaires were completed by South China citizens during the early stage of the COVID-19 pandemic based on psychological stress theory and using a comprehensive sampling method. Through cooperation with experts from other institutions, the content of the questionnaire was designed to include interviewees’ spatial locations and individual information, identify whether negative emotions were generated, and determine the level of psychological stress and the degree of perception change, etc. According to the data type, mathematical statistics and multiple logistic regression methods were used to examine regional differentiation and influencing factors regarding the psychological stress of residents using 1668 valid questionnaires from 53 municipal administrative units in South China. The results firstly showed that over the whole area there was typical regional differentiation in South China, especially in relation to negative expression and psychological stress, with this feature reflecting the dual urban–rural structure. Secondly, regional differences were obvious. Residents of Hainan showed stronger change of psychological stress than those of the other two provinces. In contrast, Guangdong residents were the least psychological stress, and the concept of a harmonious relationship between human beings and nature was not accepted as well as in the other two provinces. Thirdly, in each province the capital city acted as the regional pole, with greater psychological status. This polarization effect decreased with greater distance, reflecting the theory of growth poles in human geography. Fourthly, gender, education level, occupation, informational correction, and the possibility of infection were notable factors that affected the psychological status of interviewees facing COVID-19. However, the functions were different and were decided by the dependent variable. Lastly, based on conclusions summarized from three perspectives, it was found that regional differentiation, public information, and social structure need to focused upon in order to handle sudden major health issues.

## 1. Introduction

Major public health emergencies are characterized by suddenness, uncertainty, and gregariousness. These emergencies directly damage physical health and affect public and individual mental health [1], inducing psychological crises and sudden psychological changes [2,3]. They can even lead to extreme social and political events [4,5,6]. The COVID-19 pandemic, which began at the end of 2019, is a major public health emergency that has presented the fastest spread, the widest range of infections, and the most complex prevention and control issues since the founding of the People’s Republic of China [7]. The pandemic has led to such economic and social phenomena as supply shortages, rising prices, and the suspension of work and production, resulting in a strong psychological response from the public [8,9]. On the one hand, uncertainly around COVID-19 and the pandemic has greatly reduced the happiness of residents. An information overload regarding the pandemic and the spread of rumors have added additional psychological stress [10,11], causing anxiety, fear, and even anger [12]. On the other hand, shortages of protective materials such as masks and medicines due to the pandemic have intensified public fear and increased sensitivity to social risks [13,14]. Therefore, a scientific understanding of the impact of COVID-19 on residents’ psychological status is a basic requirement for maintaining health and mental status, stabilizing daily mental status, and reducing psychological impacts. A good understanding is also important for encouraging national spirit in the fight against the pandemic and promoting social stability and order, which are key to improving medical services and responding to public health emergencies [15].

Early research on psychological stress separated environmental conditions and individual characteristics. Psychological stress was believed to be constant and affected either by environmental or individual factors. Most studies ignored the relationship between places and people, paying more attention to the superficial phenomenon and the characteristics of the group. This approach led to such issues as the separation of the psychological stress response from the location of the incident, the limitation of individual perception and experience, and a lack of consideration of critical factors in the causal explanation of psychological phenomena [16]. In 1980s, Lazarus and Folkman proposed the psychological stress theory. They discussed how stress was a process that changed with environmental conditions and time in a state of dynamic correlation [17]. According to the Cognitive Phenomenological–Transactional Model (CPT), psychological stress is a state of physical and mental tension caused by an imbalance of the body or mind which is reflected in the emotional state, level of psychological stress, cognitive experience, and adaptive ability, etc. [18]. In CPT (Figure 1), “stressors” arise from a collection of social, cultural, physiological, and psychological factors such as major events, diseases, and cognition levels [19]. “Cognitive evaluation” and “coping” are interactive variables. In the primary evaluation, individuals use existing personal experiences and knowledge to judge the situation, while in the secondary evaluation, the individual considers what can be done to prevent and overcome harm. As the key to explaining the stress response, “coping” refers to an internal need and external behaviors when the individual makes efforts to manage stress, and generally results in two patterns of emotion and behavior. Depending on the selection, the coping pattern will change with time, the environment, and personal experiences [20]. As described above, stress generation is due to a combination of contextual factors such as environmental stimuli and the individual’s perceived threat from the environment. The interaction between people and the environment is dynamic, is associated with mental health, and can be assessed by the individual’s reactions.

As Lazarus and Folkman [7] as well as Yang Zhaoming [21] discussed, when the individual perceives the situation as being controllable, coping strategies are predominantly problem-based, while when individual cannot change the situation, emotion-based coping will occur. The response varies with the situation, location, time, and personality. The consequences are significant and have been verified after major disasters. Major public health emergencies result in specific space-based and place-based scenarios, and individuals generate perceptions regarding their location through personal experience and reflection. For example, communities closer to the New York Hurricane Center showed stronger psychological stress responses, a phenomenon which was attenuated with distance within a specific geographical area [22]. The psychological stress response level in Wuhan led to higher rates of post-traumatic stress disorder than in other areas in Hubei Province [10]. The stress coping strategies of air crash respondents are different in different regions [23]. Studies have revealed that the psychological stress levels of residents in response to major emergencies at a large scale have some common features and the interaction between individuals and environment is strong; however, cognition and coping are not only affected by the emergency itself but also by individual factors such as age, gender, occupation, and experiences [24,25]. Both the similarities and differences reflect the individuals’ evaluation, cognition, and response. Thus, psychological stress analysis from the perspective of human geography can help to record specific time–space situations, understand the cognitive coping transaction process as the event develops, and explore the spatial effect and structure of residents’ psychological stress [26,27,28].

Like other major emergencies, COVID-19 has characteristics of suddenness, spread, and regionality [29]. The separation of relatives and the implementation of self-quarantine and housing closures may act as stressors, causing varying degrees of psychological stress responses such as anxiety, panic, and stress [30,31,32,33]. Moreover, the impact of COVID-19 on the psychological state of the residents shows obvious spatio-temporal differences that are deeply affected by individual factors, government measures, situational information, and news media [34,35,36]. Relevant achievements have focused on the medicine, pathology, and epidemiology of COVID-19 [37], as well as its spatio-temporal evolution, prevention and control, and socio-economic impacts [29,38,39], ignoring the differences in psychological impacts in different spaces and among distinct groups, and restricting the interaction between residents and environment in the coping process [30]. Therefore, based on the psychological stress theory, this article used a structured questionnaire to conduct online statistical surveys with residents in South China during the period of pandemic prevention and lockdown (20–28 February 2020) as an example, and aimed to respond to several questions. Firstly, did the pandemic cause stress, and what were the spatial similarities or differences in psychological stress responses? Secondly, how did residents perceive and evaluate the pandemic and what were the influencing factors? Thirdly, how did cognition and evaluation interact in the stress coping process during the early stage of COVID-19? At the same time, this study can be regarded as an important way to show “social facts” reflecting practical problems in terms of public safety and health incidents regarding prevention and control as well as regional management and the use of psychological counseling. It is also expected to provide references for scientifically dealing with changes in the external environment, firmly establishing a safe development concept to safeguard the health and safety of residents.

## 2. Methodologies

### 2.1. Study Settings

South China is one of the most well-developed metropolitan areas in China, with a convenient transportation system, a strong mobile population, and rapid economic development. Frequent socio-economic exchanges with Central China aggravated the spread the pandemic, and most regular economic status was seriously affected. Furthermore, compulsory measures such as community lockdown, delayed school openings, and work suspension disrupted residents’ daily lives. Guangdong Province in South China has the largest population and the highest GDP in China. The Guangxi Zhuang Autonomous Region is close to Guangdong Province and is one of the China’s ethnic minority autonomous regions. Hainan Province is located in southernmost China. The island environment is relatively independent and depends on tourism. It is clear that each region is geographically and economically interconnected and has its own characteristics, which may enrich research on the impact of the pandemic among different geographical regions. In terms of the expansion of the pandemic, regarding the cumulative number of confirmed cases in China after the lockdown of Wuhan from 24 January 2020 to 26 February 2020, Guangdong Province (1347) ranked second after Hubei Province, while the Guangxi Zhuang Autonomous Region (252) and Hainan Province (168) were in the middle and last place, respectively. The selected regions were affected by COVID-19 at different levels [39,40]. Therefore, this article was focused on a survey in South China and was limited to 53 municipal administrative units in Hainan Province, the Guangxi Zhuang Autonomous Region, and Guangdong Province (excluding Hong Kong, Macau, and Sansha).

### 2.2. Questionnaire Design

A structured questionnaire for data collection was developed due to the local governance of the pandemic. Drawing on the research regarding the psychological stress framework and measurement [41,42,43], the questionnaire contained 4 main parts: (1) interviewees’ basic information, (2) daily status at home, (3) the impact of pandemic on individuals, and (4) factors that influence psychological stress. Considering obstacles such as ethnic and cultural differences, local pandemic policies, the protection of infected individuals or contacts, and sampling coverage in every region type, both the content and the form were designed to be concise and brief. With the cooperation of the experts from Chinese Academy of Sciences, the “Questionnaire of residents’ living behavior and psychological status during the COVID-19 pandemic” was established after a small-scale test. The network IP address was limited so that only 1 valid questionnaire could be completed. The interviewees could freely choose when to fill in the questionnaire, and all questionnaires were filled out anonymously, fully guaranteeing the personal privacy and authenticity of the information on the interviewees.

Relevant studies have revealed that the stress transaction process involves stressors, cognition, evaluation, and coping. This paper considered negative emotion, psychological stress, perception change, and psychological crisis as the 4 aspects of the psychological stress response, comprehensively reflecting the residents’ psychological status [1,10,17,21]. Gender, age, education level, residence, possibility of infection, fixed income, occupation, information reliability, etc. were analyzed as independent variables to examine the relationships between different psychological dimensions and factors.

In this paper, “residents’ psychological status” refer to the residents’ psychological stress process, including the 4 dimensions of negative emotion, psychological stress, perception change, and psychological crisis. The questions were set as follows:(1)Negative emotion. This dimensional question was set as “Do you feel anxious or depressed in the face of the severity of COVID-19?” (Q1), including 2 options: “yes” and “no”.(2)Psychological stress. This dimensional question was set as “If the maximum value of psychological stress is 100, what is your level of psychological stress due to COVID-19? (Q2), including 4 options: 0~30, 30~50, 50~80, and 80~100, which represent the stress ranges of very low stress, low stress, high stress, and very high stress, respectively.(3)Perception change. This dimensional question was set as “How has your perception of ‘human beings and nature in harmony’ changed as compared to before?” (Q3), including 4 options: decreased, no change, increased, and significantly increased.(4)Psychological crisis. This dimensional question was set as “Will you need psychological counseling when the pandemic is over?” (Q4), including 2 options: “yes” and “no”. If the interviewees chose “yes”, they needed to answer the question: “What kind of psychological counseling would you prefer?” (Q5), choosing at most 3 out of 5 options: “personal self-action psychological counseling”, “personal psychological counseling under professional guidance”, “collective communication counseling with homogeneous groups”, “communication counseling based on society and family relationship”, and “other types of counseling methods such as participating in religious status”.

### 2.3. Data Collection

From the end of January 2020, to stop the spread of COVID-19 and curb nationwide spread, all parts of China successively launched their first-level responses to public health emergencies and adopted comprehensive, strict, and thorough prevention and control measures. Although the pandemic situation was different among the provinces, almost all residents faced the same policy that kept them in quarantine at home, with the same sources of information from the Internet during the management and control stage of COVID-19 [44]. Online communication became a popular way for residents to communicate and could be used to conduct statistical investigations on residents’ psychological status conveniently.

This study adopted a comprehensive sampling method for data collection. Firstly, according to calculation standard of stratified random sampling, the number of samples of each province needed to be ≥288 for a 95% confidence level, 0.10 error margin, and 50% estimated answer rate, and the number of samples from the countryside, towns, urban suburbs and urban centers needed to be ≥192. Secondly, the authors established contact with local universities, research institutions, and the government. After consultations regarding the pandemic and cultural differences and with approval from the regions, a small-scale questionnaire test was conducted to guarantee the questionnaire’s accuracy. Thirdly, based on the previous procedure, a non-probabilistic snowball method was adopted to search for new interviewees using the social relationships of local people and the mini programs on Wechat, Weibo, QQ, and other online platforms.

The technical staff monitored the whole progress in the background and adjusted the sample number promptly to ensure that residents of different ages and occupations were covered. All the questionnaires were coded and examined: for example, “emotion change every 4 weeks from 20 January to 17 February” was set as a logical test item for Q1. If resident chose “yes” in Q1 and did not show negative mood in the test items, the questionnaire was considered invalid. Finally, a total of 1723 samples were collected from 20 to 28 February 2020, among which 1668 were valid (96.8%), including 533 valid questionnaires from Hainan, 666 from Guangdong, and 469 from Guangxi.

### 2.4. Sample Description

By analyzing the sample attributes of the valid questionnaires (Table 1), several observations can be made. (1) The ratio of males to females was about 4 to 6. (2) The age of the interviewees mainly was concentrated in 20–39 year range; only 4.3% of the respondents were over 50 years old. (3) The percentages of interviewees with fixed and non-fixed incomes were 51.44% and 48.56%, respectively. (4) The percentages of interviewees living in rural areas, towns, urban suburbs, and urban centers were 29.68%, 19.54%, 13.67%, and 37.11%, respectively. (5) Over 90% of the interviewees were educated to college level or above. (6) In total, 77.4% residents worked in enterprises and institutions or were students. (7) In total, 74.28% of the respondents worked in the public sector or were unemployed. In general, the samples of residents covered all industries and ages, but the respondents mainly worked in scientific research institutions and social-related public departments. This was because of the online questionnaire methodology and the social relationships of the researchers. The use of online communication platforms was also limited by age, employment, education, and other factors. However, descriptive statistics can reflect the basic characteristics of the interviewees.

### 2.5. Methods

#### 2.5.1. Contingency Table and the Chi-Squared Test

Contingency table analysis is a method that converts frequency sample variables into a second-order or multi-order cross-contingency table to test whether the row and column variables are independent, and then calculates the strength and weakness relationship between the variables. Because the row and column variables are not continuous or equidistant in contingency table analysis, they do not meet the prerequisites for simple calculation of the correlation coefficient, and the chi-squared test is generally used for judgement. The formula is as following:(1)$$\chi^{2}=\sum \frac{{\left(f_{0}-f_{e}\right)}^{2}}{f_{e}}$$
where χ^2^ is the chi-squared test value, *f_0_* is the actual observation frequency, and *fe* is the expected frequency [45].

#### 2.5.2. Multiple Logistic Regression Analysis

Multiple logistic regression is a probabilistic nonlinear regression method which is a model that estimates the arbitrary relationship between the dependent variables and the independent variables. It is mainly used to examine the correlation between sub-type variables, and can avoid the limitations of linear and curvilinear regression. Generally, one type of categorical variables is used as a reference to perform regression calculations on other types of variables.

The formula is as follows:(2)$$P\left(Y=i|x,\omega \right)=\frac{e^{\omega_{i}x}}{\left(1+\sum_{i=1}^{K-1}e^{\omega_{i}x}\right)}$$
where *P(Y = i|x, ω)* refers to the probability value corresponding to the dependent variable *Y*, *i* represents the category value of the dependent variable *Y*, *k* is the number of independent variables *x*, and *ω* is the estimated regression coefficient [46].

The dependent and independent variables in the article are categorical variables. Firstly, the last item was set for the independent variables and the first item was set for the dependent variables as the reference variable. Secondly, the chi-squared test and the likelihood ratio test were used to judge whether the various dimensions of residents’ psychological status had different distribution characteristics and relevance among the classification layers of the independent variables. Finally, the correlation between the dependent variables and the independent variables was simulated using multiple logistic regression, explaining the influence of the factors on the psychological status of the residents.

## 3. Regional Differentiation of Residents’ Psychological Status

### 3.1. Rural–Urban Differentiation

Statistical analysis of the valid questionnaires showed that the differences in negative emotions, psychological stress, and perception change in the residents living in the countryside and urban centers were greater than those in residents of towns and urban suburbs, reflecting obvious characteristics of the “dual” urban–rural structure.

(1)Negative emotion. Among the valid questionnaires, 61.63% of residents did not experience negative emotions due to the pandemic, and only 38.37% of residents felt anxious or depressed by the pandemic. The proportions of residents with anxiety or depression in the countryside, towns, urban suburbs, and urban centers were 41.01%, 36.50%, 32.01%, and 39.58%, respectively (Figure 2a), showing a “high on both sides, low in the middle” tendency for the countryside/urban centers and town/urban suburbs.(2)Psychological stress. The proportions of the residents that assessed their level of psychological stress as being very low, low, high, and very high were 18.35%, 27.28%, 32.31%, and 22.03%, respectively, with those considering themselves to be under high and very high stress representing 54.34% of respondents. Thus, over half of the residents were under greater psychological stress. Residents from different geographical areas (countryside, towns, urban suburbs, and urban centers) showed the same distribution characteristics (Figure 2b). In particular, the psychological stress of residents in the countryside was relatively high, with 56.57% being under high and very high stress, followed by the residents of towns, urban suburbs, and urban centers (56.13%, 55.26%, and 51.37%, respectively). This indicated that psychological stress gradually increased from urban centers to the countryside.(3)Perception change. The outbreak of COVID-19 profoundly affected the “eating wild game animal” behavior of ordinary people, with a reconsideration of the relationship between human beings and nature. The statistical analysis showed that the percentages of interviewees with increased and significantly increased perceptions regarding “human beings and nature in harmony” than before were 25.42% and 52.04%, respectively, accounting for 77.46% of the total interviewees. Only 2.58% and 19.96% of the interviewees showed an attenuation in this perception or no change (Figure 2c). Residents in towns showed the most obvious perception change, with 80.06% of the residents being in increased agreement with the concept of “human beings and nature in harmony”, followed by the residents of the countryside. The proportion that showed a change in perception to increased or significantly increased was 76.36%, a value lower than that of the overall samples, but higher than that for the urban centers and urban suburbs.(4)Psychological crisis. The statistical analysis showed that 94.30% of the residents in southern China did not experience a psychological crisis (Figure 2d), Only 5.70% of residents needed psychological counseling, and more than 80% of the residents who had a psychological crisis chose to resolve it through personal self-action or communication with the general population, family members, and friends. The proportions of residents in the 4 types of regions that did not require psychological counseling were all over 90%, while the percentages of residents in the countryside and towns who did experience a psychological crisis were 7.07% and 7.67% respectively, values that were 2 and 3 times greater than for those from urban centers and urban suburbs, respectively.

**Figure 2 ijerph-18-11995-f002:**
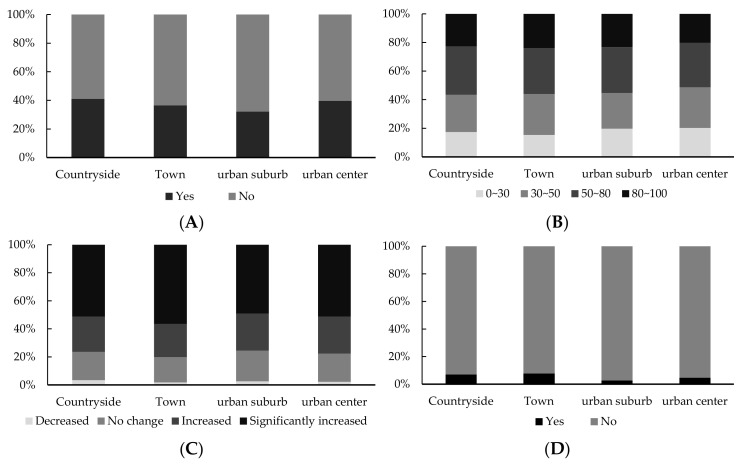
Psychological status in different areas. (**A**) Negative emotion (Q1); (**B**) Psychological stress (Q2); (**C**) Perception change (Q3); (**D**) Psychological crisis (Q4).

### 3.2. Inter-Provincial Differentiation

A statistical analysis was performed in relation to the residents’ psychological status in each province using the continency table method (Table 2).

Analyzing Table 2, it can be seen that:(1)Negative emotions. Residents feeling anxious or depressed accounted for 40.90% of the samples in Hainan Province, a value higher than that reported by residents of the Guangxi Zhuang Autonomous Region (39.66%) and Guangdong Province (35.44%).(2)Psychological stress. The psychological stress values of the residents in Hainan Province, Guangdong Province, and Guangxi Zhuang Autonomous Region were mainly concentrated in the range of 30~80, and the proportion of the residents under very high stress (80~100) was slightly greater than that relating to very low stress (0~30), indicating there are certain similar distribution characteristics of psychological stress among the provinces. Moreover, the residents of Hainan Province showed a relatively large difference in psychological stress as compared to the other two provinces, with a gap of 6.38% between residents with very high stress and very low stress, followed by the Guangxi Zhuang Autonomous Region (3.84%). Guangdong Province had the smallest gap (1.50%).(3)Perception change. One of the most fundamental changes caused by COVID-19 was the perception change in the residents. The basic principles of mutual restriction and a harmonious coexistence of “human and nature” were recognized by most residents. The statistical analysis showed that the residents of Hainan Province had the strongest sense of agreement with this perception, with 57.41% selecting the option of “very strong”, followed by residents of the Guangxi Zhuang Autonomous Region (55.44%), and residents of Guangdong Province (45.35%). The percentages of residents that selected the option of “decreased” or “no change” in perception in Guangdong Province, Guangxi Zhuang Autonomous Region, and Hainan Province were 25.97%, 20.47%, and 20.45%, respectively.(4)Psychological crisis. The survey results show that most of the residents in South China did not experience a psychological crisis. Only 5.70% residents needed psychological counseling. The proportion of the residents needing psychological counseling in Hainan Province was 6.38%, a value slightly higher than that of Guangdong Province (5.71%) and the Guangxi Zhuang Autonomous Region (4.90%).

### 3.3. Regional Differentiation

To ensure full coverage of the research unit, taking the 53 prefecture-level cities as the basic spatial unit, the spatial differentiation characteristics of psychological status in South China were analyzed by layering and coloring the proportions of residents of each dimension within the prefecture-level cities with regard to the total sample in South China (Figure 3). Moreover, the frequencies of psychological status were divided based on the distribution of percentages, with 0 indicating the absence of the psychological status, 0~0.5 referring to a lower frequency level, 0.5~1.0 expressing a low frequency level, 1.0~2.0 referring to high frequency level, and >2.0 referring to a higher frequency level. By analyzing Figure 2 it can be seen that there is a certain structural law between the spatial change trend and the frequency of psychological status on the whole, showing that some of the psychological status diffused diminishingly along the axis or periphery of regional “growth poles” such as Haikou, Guangzhou, Shenzhen, and Nanning.

(1)Negative emotion. The psychological status of residents in this dimension was relatively prominent, and the cities of Haikou, Guangzhou, Shenzhen, and Nanning showed a high frequency of occurrence regardless of whether the residents generated negative emotions or not. This phenomenon showed diffusion characteristics from the “growth pole” outwards in terms of spatial distribution (Figure 3a). However, the diffusion characteristics were different among the three provinces. The high-frequency areas in Hainan Province gradually extended from Haikou to both sides of the coast, while frequencies in Guangdong Province decreased from Guangzhou to its surrounding areas. The high-frequency areas in the Guangxi Zhuang Autonomous Region extended from Nanning to the areas adjacent to Guangdong Province.(2)Psychological stress. The psychological stress due to COVID-19 was mainly concentrated in the range of 30~80 in South China. The spatial distribution of the residents took on the characteristic of changing from weak to strong and then weak again with increased levels of psychological stress (Figure 3b). Such cities as Guangzhou, Shenzhen, and Haikou showed high frequencies of occurrence in different ranges of psychological stress. However, the spatial proximity relationship of residents’ psychological stress in other regions was not significant to their provincial capital cities, except for Guangzhou, Shenzhen, Dongguan, Zhuhai, and Huizhou, which showed regional agglomeration. Moreover, the spatial hierarchy differences in psychological stress in the range of 50~80 were the most obvious in South China.(3)Perception change. Visualizing the survey results regarding the sense of agreement with the perception that “human beings and nature in harmony” (Figure 3c), it can be seen that the stronger the sense of agreement with this perception, the more obvious the distribution characteristics of spatial aggregation. There was no spatial regularity in the spatial distribution of residents with a decreased sense of agreement, while the spatial distribution of the residents with no change or an increased or significantly increased sense of agreement showed a polarization effect centered on their provincial capital cities. For residents with a significantly increased sense of agreement in Hainan Province and the Guangxi Zhuang Autonomous Region there was a point-axis distribution centered on Nanning and Haikou which extended along the axis of the respective surrounding cities, while Guangdong Province showed a gradually weakened extension from the multiple poles of Guangzhou, Shenzhen, and Zhuhai to peripheral areas such as Foshan, Huizhou, and Dongguan.(4)Psychological crisis. As opposed to negative emotion, the spatial distribution of residents that needed psychological counseling showed a weak correlation with the provincial capital cities, with no characteristics of spatial hierarchy (Figure 3d). Moreover, the 12 cities of Guilin, Nanning, Guigang, Yulin, Guangzhou, Foshan, Dongguan, Shenzhen, Zhuhai, Haikou, Danzhou, and Sanya showed a high frequency of areas with residents who did not experience a psychological crisis. There were only 11 cities that were low-frequency areas in this regard. This indicated that residents of most cities in South China did not experience psychological crisis, in line with the statistical analysis results of the above-mentioned regions.

All in all, residents’ psychological status in South China showed geographic differentiation at different urban–rural, inter-province, and regional levels in relation to COVID-19. It is not necessary to consider the interaction among the three scales for their mutual testing.

**Figure 3 ijerph-18-11995-f003:**
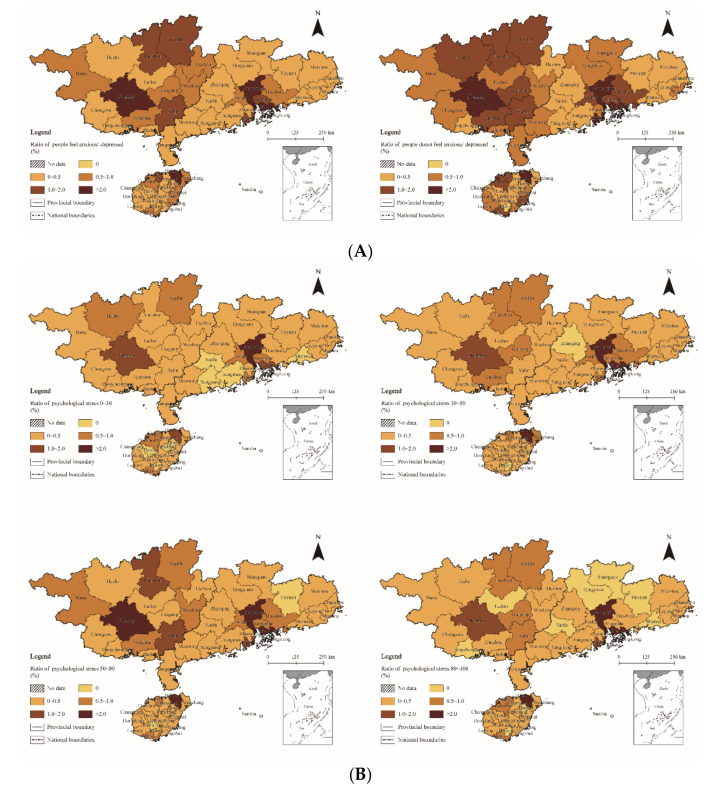
Spatial distribution of psychological status ratios. (**A**) Negative emotion (Q1); (**B**) Psychological stress (Q2); (**C**) Perception change (Q3); (**D**) Psychological crisis (Q4).

## 4. Influencing Factor Analysis of Residents’ Psychological Status

### 4.1. Model Selection and Testing

The multiple logistic regression model was used to analyze influencing factors of psychological status in South China, and used 4 dimensions of psychological status (negative emotion, psychological stress, perception change, and psychological crisis) as the stratified dependent variables. Gender, age, education level, fixed income, nature of employment, occupation, information reliability, and the possibility of infection were the independent variables. If the value of the chi-squared significance test is > 0.05, it means that there is no difference between the dependent variable and the independent variable layers. If the value of the chi-squared significance test is ≤ 0.05, this means that there is a significant difference in the spatial distribution between the dependent variables and the independent variable layers. Moreover, based on the difference between the layers, it can further be determined whether there is a correlation through the significance level of the likelihood ratio. If the value of the likelihood ratio test is ≤0.05, this indicates that the independent variables and the dependent variable are correlated (Table 3).

(1)Negative emotion. The values of the chi-squared test of variables such as gender, age, fixed income, nature of employment, occupation, and possibility of infection were ≤0.05, indicating that the spatial distribution of negative emotions showed a significant difference in the layers of the above independent variables. However, the spatial distribution of negative emotions was homogenous in the layers of education level and information reliability. Furthermore, the variables of gender and possibility of infection passed the likelihood ratio test and were correlated with negative emotions.(2)Psychological stress. The spatial distribution of psychological stress showed significant differences in the layers of seven independent variables besides the education level. Moreover, the values of the likelihood ratio tests of gender, information reliability, and possibility of infection were ≤0.05, indicating that they had statistically significant relationships with psychological stress.(3)Perception change. The values of the chi-squared tests of gender, education level, information reliability, and possibility of infection were ≤0.05, indicating that the spatial distribution of perception change showed significant differences in the layers of the above independent variables. Moreover, the values of the likelihood ratio test of gender, education level, and information reliability passed the likelihood ratio test and were correlated with perception change. However, factors such as age, fixed income, nature of employment, and occupation did not show a statistically significant correlation with perception change.(4)Psychological crisis. The factors of education level, occupation, information reliability, and possibility of infection passed the chi-squared test and likelihood ratio test, respectively, indicating that the spatial distribution of psychological crisis showed significant differences in the layers of the above independent variables. However, the psychological crisis variables showed a relatively weak correlation with the factors of gender, age, fixed income, and nature of employment.

### 4.2. Multiple Logistic Regression Analysis

Multiple logistic regression analysis was conducted on factors with significant differences and correlations in the layers of the various dimensions of psychological status to determine the degree of influence of different types of factors on residents’ psychological status in South China based on the chi-squared test and the likelihood ratio test. The results are shown in Table 4, where *B* is the regression coefficient and *Exp(B)* is the multiple value of the occurrence probability of a certain independent variable with regard to the other corresponding categories.

(1)Negative emotion. A significant positive correlation with no negative psychological emotions was shown in males as compared to females, with a coefficient of 0.28, indicating that the ratio of males that had non-negative emotions to those that had negative emotions was 1.32 times that of females. That is, females were more likely to feel anxiety or depression due to the pandemic. The chi-squared test and likelihood ratio test indicated that the possibility of infection factor was related to negative emotions, while the results of regression analysis further showed that the category of possibility of infection showed no differences regarding the impact of negative emotions.(2)Psychological stress. A significant negative correlation with psychological stress in the range of 30~80 was found in males as compared to females. The incidence of psychological stress in males in the range of 30~50 was 0.61 times that of females, decreasing to 0.57 times in the range of 50~80, indicating that females can withstand greater psychological stress than males in the range of 30~80. However, gender had an indistinguishable influence on psychological stress in the range of 80~100. Besides, the possibility of infection in the residents themselves had a significantly positive correlation with psychological stress, indicating that psychological stress gradually strengthened with the increasing possibility of infection. This was prominently shown in the range of very high psychological stress, indicating that viral infection was the main cause of psychological stress.(3)Perception change. The incidence values of perception change in males with regard to the relationship between human beings and nature from none, strong, to very strong were 0.49 times, 0.48 times, and 0.33 times those of females, respectively, indicating that the sense of agreement with this perception in females was stronger than that in males. The gender gap gradually strengthened with the increased sense of agreement. In terms of information reliability, there was a significant negative correlation with unchanged perceptions when the residents believed that the pandemic information was 100% true. However, the significant relationship changed from negative to positive and gradually increased along with the decrease in information reliability during the pandemic. For example, when the information reliability decreased to 80~100%, the incidence of those with a increased and significantly increased sense of agreement with this concept was 15.05 and 9.03 times that of those with a decreased sense of agreement, respectively, indicating that the sense of agreement regarding the relationship between human beings and nature increased with decreased information reliability. Compared with high-level education, residents with elementary school-level education or below were significantly and positively correlated with no change and significantly increased of perception change, indicating an obvious polarized phenomenon of percaption change among the resicents with low-level education.(4)Psychological crisis. The incidence of no psychological crises in residents with a junior high school and senior high school-level education was 0.36 times and 0.42 times that of residents with a college degree-level education or above, respectively, indicating that those with a college degree or above level of education were more likely to experience a psychological crisis and need psychological counseling. Therefore, residents with an elementary school-level education or below were significantly less likely to experience a psychological crisis. Residents who believed that they had no, little, or an uncertain possibility of infection were more likely to not experience psychological crises. Furthermore, students were significantly less likely to experience a psychological crisis as compared to those of other occupations.

## 5. Discussion

During the home quarantine period due to the COVID-19 pandemic, residents in South China showed different levels of psychological stress, although they were mainly concentrated in the range of 30~80. With the transition in location from urban to rural areas, the average stress gradually decreased. The spatial hierarchies of psychological stress changed from weak to strong and then weak, and only residents of mega-cities like Guangzhou, Shenzhen, and Dongguan showed very high levels of psychological stress and high-level agglomeration. The causes of such stress was clearly related to the residents’ possibility of infection. As noted by other researchers [41,42,47], the spread of the pandemic was one of the main causes of psychological stress. It is worth noting that females seemed to be more stressed than males.

Lazarus and Folkman’s theory explained that both cognition and evaluation are closely related to stressors, environment, and individuals [17,18]. This study considered changes in residents’ perceptions of the relationship between human beings and nature to represent people’s cognition. In the evaluation, the residents in both Hainan Province and the Guangxi Zhuang Autonomous Region displayed a stronger sense of change than those of Guangdong Province. The spatial differences in perception change in the residents were strengthened by an increased sense of agreement with this concept, and decreased from the provincial capital cities to the surrounding areas of each province. Residents’ evaluations were significantly correlated with gender, education level, and pandemic information reliability. For example, in interviewees who were more highly educated or believed that over 80% of information was reliable, the cognitive process was stable and perceptions basically unchanged.

Moreover, in terms of coping strategies, approximately 38.7% residents showed negative emotions due to the pandemic. As with perception change, negative emotion showed typical urban–rural geographical distribution characteristics of “high on both sides and low in the middle”. The incidence of negative emotions in Hainan Province was higher than in Guangdong Province and the Guangxi Zhuang Autonomous Region. The spatial distribution of negative emotions decreased from the regional center cities of the provinces of Haikou, Guangzhou, Shenzhen, and Nanning to their outside areas, respectively. On the other hand, most surveyed residents did not experience a psychological crisis and did not need psychological counseling. Meanwhile, the percentage of residents that required psychological counseling who were from countryside or urban center samples was more than three times that of those from urban suburbs. The incidence of psychological crisis in the residents of Hainan was higher than that of residents from Guangdong Province and the Guangxi Zhuang Autonomous Region. The spatial distribution of residents who experienced a psychological crisis was relatively weakly correlated with the polarization of the provincial capitals. Among the influencing factors, education level, occupation, and possibility of infection were significantly correlated with psychological crises.

The results of the study showed that residents experienced different levels of stress. The differences in cognitive values and evaluations were influenced by the development of the pandemic, individual education levels, and gender. There were also different characteristics in terms of coping strategies. Although the differential features in each part were geographically and mathematically clear, changes in cognitive evaluations under different stress levels had a significant predictive effect with regard to coping response. For example, people who showed high levels of stress tended to experience negative emotions or even psychological crises. Influencing elements such as gender, possibility of infection, pandemic information, and educational level contributed to psychological responses to different extents, reflecting the transactional nature of stress processes that are determined not only by contextual factors but also by personal factors. From a geographical perspective, the spatio-temporal pattern of residents’ psychological status provides us with an in-depth understanding of the interaction between human beings and the environment, and to some extent policies and governance should be more focused on urban–rural duality and the urban polarization effect.

## 6. Conclusions

COVID-19 is a major global public health emergency. This study attempted to construct a quantitative table of psychological status to deductively analyze regional differentiation and influencing factors of residents’ psychological stress in South China in the initial stages of control of the pandemic. Due to the impact of the pandemic itself and lockdown, it was difficult to use the traditional stress questionnaire during the first outbreak of COVID-19. Instead of using internationally validated psychological scales, this study adopted a structured questionnaire that was designed to consider regional cultural, social, and pandemic-related differences to collect data online. The results showed that residents’ psychological stress had space-based and people-based characteristics. However, the questionnaire cannot fully describe the mechanisms and processes, and perhaps there is a certain degree of subjectivity in terms of specific operations. In future studies a validated questionnaire could be used to conduct interviews, address behavioral observations, and enrich multi-disciplinary methods to verify the transactional process and explain the psychological phenomenon more clearly. Moreover, Lazarus and Folkman noted that stress is an adaptive process. The authors collected data from 21 to 28 February 2020, when residents had experienced the outbreak of the pandemic, city lockdown, isolation prevention, and work and school out of orders. The analysis showed how residents responded to COVID-19 in the early period. However, the pandemic is still affecting residents’ lives, and psychological stress status is changing accordingly. Thus, a longer-term survey is needed to provide a clear picture of the impact of COVID-19. The authors will continue to focus on the influence of the pandemic and continue the investigation of residents’ psychological status over multiple periods. Lastly, the intention of this study was to provide a discussion on the psychological impact of COVID-19 in South China. It remains a highly geographical phenomenon that is related to diverse local properties as well as cultural, economic, and political factors. As such, the findings may not be fully applicable to other settings.

The results have great practical significance for understanding the spatio-temporal differentiation of psychological status, strengthening the prevention and control of the pandemic, and aiding in the recovery of psychological health. Therefore, some suggestions are put forward. Firstly, more attention should be paid to the “dual” structure of urban–rural areas and the influence of the “growth pole” of provincial capitals. The prevention and control of the pandemic and joint control management approaches need to focus on the “top-down” proliferation effect of the cities. However, differences in regional types should be considered in the implementation of specific measures, with “bottom-up” measures to scientifically guide rural residents to respond to major emergencies. Secondly, specific groups, such as females and those with lower levels of education, need to receive more accurate, professional, and authoritative help. Thirdly, it is necessary to popularize public scientific knowledge to eliminate panic and improve cognitive ability. In the meantime, the government should build positive images to enhance confidence in defeating the pandemic.

## Figures and Tables

**Figure 1 ijerph-18-11995-f001:**
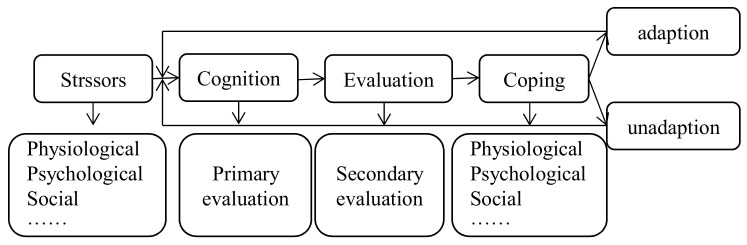
The Cognitive Phenomenological–Transactional Model (CPT).

**Table 1 ijerph-18-11995-t001:** Descriptive statistics of the samples.

Variable Name	Variable Assignment	N	%
Gender	1 = male	684	41.01
	2 = female	984	58.99
Age	1 = under 20 years old	272	16.31
	2 = 20~29 years old	728	43.65
	3 = 30~39 years old	401	24.04
	4 = 40~49 years old	195	11.69
	5 = 50~59 years old	58	3.48
	6 = over 60 years old	14	0.84
Residence	1 = countryside	495	29.68
	2 = town	326	19.54
	3 = urban suburb	228	13.67
	4 = urban center	619	37.11
Education level	1 = primary school and below	5	0.30
	2 = junior high school	43	2.58
	3 = senior high school or technical secondary school	114	6.83
	4 = junior college and above	1506	90.29
Possibility of infection	1 = none	316	18.94
	2 = low	571	34.23
	3 = no idea	525	31.47
	4 = high	254	15.23
	5 = very high	2	0.12
Fixed income	1 = yes	858	51.44
	2 = no	810	48.56
Nature of employment	1 = no fixed unit	106	6.35
	2 = public sector	619	37.11
	3 = private sector	323	19.36
	4 = no job	620	37.17
Occupation	1 = employees of enterprises and institutions	582	34.89
	2 = middle-level and above leading cadres	147	8.81
	3 = entrepreneurs	59	3.54
	4 = students	709	42.51
	5 = farmers	12	0.72
	6 = retires	17	1.02
	7 = others	142	8.51
Information reliability	1 = 100%	232	13.91
	2 = 80~100%	817	48.98
	3 = 60~80%	526	31.53
	4 = 40~60%	73	4.38
	5 = 20~40%	14	0.84
	6 = under 20%	6	0.36

**Table 2 ijerph-18-11995-t002:** Psychological status in each province.

Dimensions	Options	Hainan Province	Guangdong Province	Guangxi Zhuang Autonomous Region
Negative emotion	Yes	40.90	35.44	39.66
	No	59.10	64.56	60.34
Psychological stress	0~30	16.32	20.27	17.91
	30~50	28.52	27.48	25.59
	50~80	32.46	30.48	34.75
	80~100	22.70	21.77	21.75
Perception change	Decrease	2.63	3.00	1.92
	No change	17.82	22.97	18.55
	Increase	22.14	28.68	24.09
	Significant increase	57.41	45.35	55.44
Psychological crisis	Yes	6.38	5.71	4.90
	No	93.62	94.29	95.10

**Table 3 ijerph-18-11995-t003:** Correlation between psychological status and influencing factors.

Variables	Negative Emotion(Q1)		Psychological Stress(Q2)		Perception Change(Q3)		Psychological Crisis(Q4)	
	*χ* ^2^	*LR*	*χ* ^2^	*LR*	*χ* ^2^	*LR*	*χ* ^2^	*LR*
Gender	0.004	0.005	0	0	0	0	0.513	0.789
Age	0.009	0.436	0	0.179	0.183	0.43	0.834	0.62
Education level	0.174	0.534	0.139	0.109	0.003	0	0	0.024
Fixed income	0.022	0.129	0.002	0.704	0.424	0.856	0.811	0.417
Nature of employment	0.017	0.795	0.021	0.533	0.1	0.329	0.491	0.93
Occupation	0.001	0.215	0.005	0.687	0.201	0.527	0	0
Information reliability	0.389	0.262	0	0.006	0	0	0	0.05
Likelihood of infection	0	0	0	0	0	0.133	0	0.002

Note: χ2 refers the value of the chi-squared test; LR means the value of likelihood ratio test.

**Table 4 ijerph-18-11995-t004:** The simulation results of multinational logistic regression.

Variable	Category	Negative Emotion(Q1)		Psychological Stress(Q2)					
		No		30~50 (Low)		50~80 (High)		80~100 (Very High)	
		*B*	*Exp(B)*	*B*	*Exp(B)*	*B*	*Exp(B)*	*B*	*Exp(B)*
Gender	1	0.279 *	1.322	−0.492 *	0.611	−0.560 *	0.571	−0.241	0.786
Possibility of Infection	1	1.12	3.066	17.543 *	4.15 × 10^7^	17.418 *	3.67 × 10^7^	−0.607	0.545
	2	1.09	2.975	17.984 *	6.46 × 10^7^	17.957 *	6.29 × 10^7^	−0.715	0.489
	3	0.435	1.546	18.494 *	1.08 × 10^8^	18.675 *	1.29 × 10^8^	−0.569	0.566
	4	−0.014	0.986	18.100 *	7.26 × 10^7^	18.810 *	1.48 × 10^8^	−0.316	0.729
Information reliability	1			−0.501	0.606	−0.4	0.67	−0.541	0.582
	2			0.927	2.527	0.981	2.666	0.665	1.944
	3			0.865	2.374	0.751	2.119	0.367	1.443
	4			1.079	2.941	0.774	2.167	0.401	1.494
	5			−0.023	0.978	0.383	1.466	−0.349	0.705
**Variable**	**Category**	**Perception Change(Q3)**						**Psychological Crisis(Q4)**	
		**No change**		**Increased**		**Significantly increased**		**No**	
		** *B* **	** *Exp(B)* **	** *B* **	** *Exp(B)* **	** *B* **	** *Exp(B)* **	** *B* **	** *Exp(B)* **
Gender	1	−0.708 *	0.493	−0.733 *	0.48	−1.124 *	0.229		
Information reliability	1	−2.335 *	0.097	−0.098	0.907	−0.835	0.434	−0.369	0.691
	2	1.196 *	3.307	2.711 *	15.048	2.201 *	9.032	0.638	1.892
	3	1.462 *	4.313	3.218 *	24.977	2.489 *	12.053	1.058	2.882
	4	2.278 *	9.754	3.181 *	24.068	2.869 *	17.617	0.714	2.041
	5	0.343	1.409	0.978	2.658	0.911	2.488	0.099	1.104
Education level	1	18.533 *	1.12 × 10^8^	−0.449	0.639	16.303 *	1.20 × 10^7^	17.442 *	3.76 × 10^7^
	2	0.225	1.252	−0.726	0.484	−0.724	0.485	−1.032 *	0.356
	3	−0.255	0.775	−0.635	0.53	−0.428	0.652	−0.870 *	0.419
Risk perception	1							2.604 *	13.511
	2							3.008 *	20.242
	3							2.709 *	15.013
	4							1.026	2.789
Occupation	1							0.541	1.717
	2							0.347	1.415
	3							0.835	2.305
	4							0.729 *	2.073
	5							−1.003	0.367
	6							18.204	8.05 × 10^7^

Note: variable items are shown in Table 1. *B* is the regression coefficient, *Exp(B)* = *eB*. * indicates a 5% significance level.

## Data Availability

The data used to support the findings of this study are available from the corresponding author upon reasonable request.

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
