# Peer review of "Regional Differentiation and Influencing Factor Analysis of Residents’ Psychological Status during the Early Stage of the COVID-19 Pandemic in South China"

_ijerph, 2021, doi:10.3390/ijerph182211995_

Round 1
Reviewer 1 Report
The manuscript describes the psychological state of the inhabitants of South China in the early moments of COVID-19. To do this, the impact of different contextual factors is analyzed. The topic addressed is of interest from a public health point of view, and the study analyzes several relevant factors in a large sample. However, in my opinion, the manuscript has serious conceptual and methodological weaknesses. These weaknesses are listed below (please, pay special attention to points 5 and 6):
- Abstract: More precise information should be provided regarding the method: participants and sampling procedure, methodological design, etc.
- The study takes Lazarus and Folkman's psychological theory of stress as a conceptual frame of reference. This theory emphasizes the eminently interactive (transactional) nature of the stress process, determined not only by contextual factors (stressors), but also by personal factors (fundamentally, the way in which the individual cognitively values the situation and copes with it). In fact, Lazarus and Folkman state that coping is the key to explain the stress response. For this reason, I believe that the authors should more clearly reflect this transactional nature of stress in their explanation.
- Since the study focuses on South China, a broader contextualization of this region should be provided. It would also be important to explain to what extent these contextual factors can pose a threat to the psychological health of the population of South China.
- The specific objectives of the investigation should be indicated with greater specificity.
- The design of the questionnaire used is poor. Some items even pose dichotomous responses, which barely provide information and may be biased. Likewise, the process followed to prepare the questionnaire should be explained: reference sources, items initially raised and eliminated, process for agreeing on the items finally used, etc.
- The data collection was carried out between February 21 and 28, 2020. It is a very short time to draw conclusions about the psychological state derived from COVID. A longer period of time is needed, as only the first moments since the onset of the pandemic have been considered. It should be taken into account that stress is an adaptive process, in which the person needs to assimilate threatening events to respond to them. Probably, those 7 days analyzed do not offer a rigorous “photograph” of the real impact of COVID.
- Line 456: Sorry to disagree, but I don't see the connection to Lazarus and Folkman's stress theory. In reality, this section lacks a true discussion of results as such. The conclusions are overly descriptive. The findings are not compared with those of other studies, nor are the results interpreted in the light of any theory (e.g., transactional stress theory).
- It is necessary to expand the explanation regarding the health implications of the study results, as well as future lines of research.
Reviewer 2 Report
This work offers insights on the Residents’ Psychological Activities During the Early Stage of COVID-19 in South China. The work is of a good standard and interesting for the readers of the journal. However, some points need to be further addressed.
- Please, clarify which "residents' psychological activities” the authors are referring to.
- Questionnaire design: why haven't validated scales been used to measure the psychological constructs? This represents a strong limitation that must be justified.
- Furthermore, has the study that intended to examine psychological variables received the approval of an ethics committee? Suggest specifying.
- Were there criteria for inclusion / exclusion from the study? How was the questionnaire distributed? Suggest including a detailed part on the procedure of data collection.
- In the method part, please mention that the demographic characteristics of the participants are collected.
- The discussion is the weakest part. It is necessary to improve it, especially in light of the huge number of studies carried out on Covid-19 pandemic. For example, the differences between rural and urban environment could be explained in many ways including the presence / absence of natural elements that have proven to affect negative emotions during the pandemic (e.g., 10.1016/j.ufug.2021.127156). The same should be done for each indicator subsequently examined.
- Lastly, it is advisable to divide the discussion and the conclusions’ part into two separate paragraphs.
Reviewer 3 Report
Dear authors,
this article aims to analyze regional differences and factors influencing the psychological response to the covid 19 pandemic in southern China residents.
The objective of identifying the psychological effects through the 4 dimensions that the authors decide to consider: negative expression, psychological pressure, change in perception and psychological crisis, is correct.
However, it is considered too simplistic and without scientific support to carry out the evaluation of each of these dimensions respectively through the formulation of a single question, mostly of a binary nature.
From a simple analysis of the literature it emerges how different scientifically recognized and approved questionnaires, such as the "Composite International Diagnostic Interview - CIDI", the "Social Phobia Inventory (SPIN)", etc.
Could have supported, in a robust way purpose that the authors had set.
Round 2
Reviewer 1 Report
The authors have done a great job and have been able to respond satisfactorily to all the questions requested. However, I think they should indicate more clearly the limitations of this study. Among others, the use of an ad hoc questionnaire (not validated) or the short period of time evaluated (7 days).
Reviewer 2 Report
I am satisfied with authors' replies to my comments, thus I endorse publication of this article in its current version.
